# Induction of p53-Dependent Apoptosis by Prostaglandin A_2_

**DOI:** 10.3390/biom10030492

**Published:** 2020-03-24

**Authors:** Su-Been Lee, Sangsun Lee, Ji-Young Park, Sun-Young Lee, Ho-Shik Kim

**Affiliations:** 1Department of Biomedicine and Health Sciences, College of Medicine, The Catholic University of Korea, Seoul 06591, Korea; tnqls9437@naver.com (S.-B.L.); fight1014@nate.com (S.L.); jpweb@catholic.ac.kr (J.-Y.P.); 2Cancer Evolution Research Center, College of Medicine, The Catholic University of Korea, Seoul 06591, Korea; 3Department of Biology, Faculty of Science, Naresuan University, Phitsanulok 65000, Thailand

**Keywords:** apoptosis, prostaglandin A_2_, tumor suppressor Protein p53, DNA-activated protein kinase, death receptor

## Abstract

Prostaglandin (PG) A_2_, one of cyclopentenone PGs, is known to induce activation of apoptosis in various cancer cells. Although PGA_2_ has been reported to cause activation of apoptosis by altering the expression of apoptosis-related genes, the role of p53, one of the most critical pro-apoptotic genes, on PGA_2_-induced apoptosis has not been clarified yet. To address this issue, we compared the apoptosis in HCT116 *p53* null cells (HCT116 p53-/-) to that in HCT116 cells containing the wild type *p53* gene. Cell death induced by PGA_2_ was associated with phosphorylation of histone H2A variant H2AX (H2AX), activation of caspase-3 and cleavage of poly(ADP-ribose) polymerase 1 in HCT116 cells. Induction of apoptosis in PGA_2_-treated cells was almost completely prevented by pretreatment with a pan-caspase inhibitor, z-VAD-Fmk, or an inhibitor of protein synthesis, cycloheximide. While PGA_2_ induced apoptosis in HCT116 cells, phosphorylation of p53 and transcriptional induction of p53-target genes such as *p21^WAF1^*, *PUMA*, *BAX*, *NOXA*, and *DR5* occurred. Besides, pretreatment of pifithrin-α (PFT-α), a chemical inhibitor of p53’s transcriptional activity, interfered with the induction of apoptosis in PGA_2_-treated HCT116 cells. Pretreatment of NU7441, a small molecule inhibitor of DNA-activated protein kinase (DNA-PK) suppressed PGA_2_-induced phosphorylation of p53 and apoptosis as well. Moreover, among target genes of p53, knockdown of *DR5* expression by RNA interference, suppressed PGA_2_-induced apoptosis. In the meanwhile, in HCT116 p53-/- cells, PGA_2_ induced apoptosis in delayed time points and with less potency. Delayed apoptosis by PGA_2_ in HCT116 p53-/- cells was also associated with phosphorylation of H2AX but was not inhibited by either PFT-α or NU7441. Collectively, these results suggest the following. PGA_2_ may induce p53-dependent apoptosis in which DNA-PK activates p53, and DR5, a transcriptional target of p53, plays a pivotal role in HCT116 cells. In contrast to apoptosis in HCT116 cells, PGA_2_ may induce apoptosis in a fashion of less potency, which is independent of p53 and DNA-PK in HCT116 p53-/- cells

## 1. Introduction

Tumor suppressor gene, p53, which is one of the most crucial tumor suppressor genes, exerts its anti-cancer effect by activating cell death, including apoptosis and autophagic cell death, as well as cell cycle arrest in cancer cells [1]. Apoptosis induced by p53 is carried out by apoptosis-relating proteins whose expression is modulated by p53. Besides, p53 can induce apoptosis by directly stimulating cytochrome c release from mitochondria through its mitochondrial translocation [2,3,4]. The intracellular level of p53 protein is negatively regulated by the ubiquitin-proteasome system (UPS). In the basal state, p53 is ubiquitinylated by MDM2, an E3 ubiquitin ligase, and then degraded in the 26S proteasome, thereby keeping the level of p53 protein at the minimum. On the other hand, genotoxic stresses such as gamma-irradiation, ultraviolet (UV) light, and oxidative stress, increase the level of p53 protein via protein kinase cascade phosphorylating p53. Since phosphorylated p53 does not bind to MDM2, p53 escapes from UPS, resulting in the upregulation of p53 protein [5,6]. Increased p53, in turn, stimulates the transcription of numerous genes, including *PUMA*, *NOXA*, and *BAX* which can trigger apoptosis as well as *p21^WAF1^*, an inhibitor of CDK4/6, and CDK2, which blocks the cell cycle progression in late G1 phase. The p53 gene is mutated in over 50% of all human cancers, and the introduction of intact p53 in cancer cells containing mutant p53 induces growth suppression and apoptosis. Therefore, p53 is considered as the most critical transcription factor that induces apoptosis in cancer cells, which is reflected by the fact that most anti-cancer therapeutics, including chemotherapy and radiotherapy, induce apoptosis by activating p53 [1,7].

Prostaglandin (PG) A_2_, one of cyclopentenone PGs (cyPGs), is produced from PGE_2_ by non-enzymatic dehydration. In contrast to PGE_2_, which acts through its cognate receptor in the cytoplasmic membrane [8,9], PGA_2_ directly transports to the nucleus, where it induces de novo synthesis of proteins on which the biological functions of PGA_2_ are dependent [8]. Many research groups have reported that PGA_2_ affects various biological processes such as differentiation of leukemic cells, cell cycle progression and induction of apoptosis in various cancer cell lines, including hepatocellular carcinoma, breast cancer, cervical cancer, and leukemia [10,11,12,13]. Treatment of PGA_2_ activates apoptosis both in caspase-dependent and -independent manners by modulating the expression of apoptosis-related proteins. For example, PGA_2_ induces the expression of BAX in MCF-7 cells [14] and SOX-4 and c-MYC in Hep3B cells, and these proteins were critical in inducing apoptosis [15,16].

However, the role of p53 in the PGA_2_-induced apoptosis in cancer cells has not been elucidated yet. In hepatoma cells, PGA_2_ increased p53 protein during the induction of apoptosis and 15-deoxy-Δ^12,14^-PGJ_2_ (15d-PGJ_2_) activated p53 via ataxia telangiectasia mutated (ATM) activation, suggesting that PGA_2_-induced apoptosis may be mediated by p53 [15,17]. On the contrary to these reports, it has been reported that PGA_2_ and 15d-PGJ_2_ inhibit the transcriptional activity of p53 through their direct binding to p53 resulting in conformational changes of p53 [18,19].

Recently, we have reported that PGA_2_ hinders cell cycle progression by activating the transcription of *GADD45a* and *HO-1*, which was initiated by oxidative stress [20]. Furthermore, we observed that inhibition of p53’s transcriptional activity prevented the expression of HO-1, and PGA_2_ treatment in cancer cells led to phosphorylation of p53, via activation of DNA-PK [21]. These findings strongly suggest the possibility that PGA_2_ may induce apoptosis via activation of p53. To address this issue, we analyzed whether PGA_2_ induces the activation of p53 during the induction of apoptosis by comparing the level of apoptosis between HCT116 cells containing the wild-typep53 gene and HCT116 p53 null isogenic (HCT116 p53 -/-) cells. Besides, the molecular mechanism involved in PGA_2_-induced apoptosis in HCT116 cells was investigated.

## 2. Materials and Methods

### 2.1. Chemicals

Prostaglandin A_2_ was obtained from Biomol International Inc. (Plymouth Meeting, PA, USA). Pifithrin-α (PFT-α), cycloheximide (CHX) and z-VAD-Fmk were purchased from Sigma-Aldrich Inc. (St. Louis, MO, USA). NU7441 was from Tocris Bioscience (Bristol, UK). All reagents used in this study were of molecular biology, or cell culture tested grade.

### 2.2. Cell Culture

Human colorectal HCT116 cells obtained from the American Type Culture Collection (ATCC) (Manassas, VA, USA) were maintained in RPMI 1640 supplemented with 10% fetal bovine serum (FBS) (Hyclone, Logan, UT, USA), 100 units/mL penicillin (Hyclone) and glutamate (Invitrogen, Carlsbad, CA, USA) at 37 °C under 5% CO_2_. The subculture of cells or replacement of media was done every three days.

### 2.3. Cell Death Assay

Cell death induced by PGA_2_ was assessed by annexin V assay. After PGA_2_ treatment, cells were stained with fluorescein isothiocyanate (FITC)-labeled annexin V to measure the translocation of phosphatidylserine from inner leaflets to outer leaflets in the cytoplasmic membranes and propidium iodide to detect necrotic nuclei using Annexin V apoptosis assay kit (BD Biosciences, San Jose, CA, USA). Then, fluorescence of stained cells was measured on FACSCanto II (BD Biosciences) and analyzed using BD FACSDiva program.

### 2.4. Immunoblot Analysis

For the detection of changes in the expression of proteins in PGA_2_-treated cells, cells were lysed in radioimmunoprecipitation assay (RIPA) buffer containing cocktails of protease inhibitors (Roche, Basel, Switzerland) and phosphatase inhibitors (FIVEphoton Biochemicals, San Diego, CA). All antibodies were commercially available as follows. Rabbit anti-p53, anti-phospho-p53 (Ser-15), DR5 (death receptor 5), and anti-active caspase-3 were purchased from Cell Signaling Technology (Boston, MA, USA). Rabbit cleaved PARP1 (c-PARP1) and chicken anti-glyceraldehyde-3-phosphate dehydrogenase (GAPDH) antibodies were obtained from Abcam (Cambridge, UK) and Merck Millipore Korea (Seoul, Korea), respectively. Peroxidase-conjugated antibodies (HRP-conjugated anti-rabbit or -mouse IgG) were from Sigma-Aldrich Inc., and KPL (Gaithersburg, MD, USA, HRP-conjugated anti-chicken IgG).

### 2.5. Quantitative Real Time Polymerase-Chain Reaction

For the analysis of the mRNA level of target genes of p53 in PGA_2_-treated cells, quantitative real time polymerase-chain reaction (qPCR) was performed as follows. First-strand cDNA was synthesized from total RNA using PrimeScript^TM^ RT reagent Kit (Takara Korea Biomedical Inc., Seoul, Korea). First-strand cDNA was then amplified by specific primers against target genes of p53 using SYBR FAST qPCR Kit (KAPAbiosystems, Woburn, MA, USA) on ABI 7300 Real-Time PCR System (Applied Biosystems, Carlsbad, CA, USA). GAPDH mRNA level normalized each mRNA level of p53-target genes in the same sample and their relative changes among samples were calculated by the ΔΔCt method [22].

### 2.6. Transfection of Small Interfering RNA (siRNA)

For the knockdown of *DR5* expression, siRNA against *DR5* was transfected into HCT116 cells using Lipofectamine^TM^ RNAiMAX Transfection reagent (Thermo Fisher Scientific, Waltham, MA, USA). The final concentration of DR5 siRNA was 1 nM, and the volume of transfection reagent was 3 µL.

### 2.7. Statistical Analysis

All data in this study are expressed as the means ± standard error of the mean, which were obtained from three independent experiments performed in triplicate. Statistical analysis was performed using a paired Student’s *t*-test. *P*-values of data were indicated in each figure.

## 3. Results

### 3.1. PGA_2_ Induces Caspase-Dependent Apoptosis in HCT116 Cells But Not in HCT116 p53-/- Cells

First, we attempted to observe the difference in the level of apoptosis between HCT116 cells and HCT116 p53-/- cells treated with PGA_2_. As shown in Figure 1A,B, and Appendix A, annexin V-positive HCT116 cells were increased according to concentrations and incubation times of PGA_2_, but in HCT116 p53-/- cells treated with PGA_2_, annexin V-positive cells were hardly increased. To be consistent with the result of annexin V assay, PGA_2_ induced cleavage of poly(ADP-ribose) polymerase 1 (PARP1) and caspase-3 only in HCT116 cells (Figure 1C). Moreover, when HCT116 cells were pretreated by z-VAD-Fmk, a pan-caspase inhibitor, PGA_2_-induced apoptosis was almost wholly prevented (Figure 1D,E). Collectively, these data suggested that PGA_2_ induces caspase-dependent apoptosis in a p53-dependent manner in HCT116 cells. 

### 3.2. PGA_2_ Activates p53 via DNA-PK during the Induction of Apoptosis in HCT116 cells

Then, we analyzed whether and how p53 was activated in HCT116 cells during the PGA_2_-induced apoptosis. Whereas HCT116 p53-/- cells showed no expression of p53, p53 was phosphorylated at Ser-15 and at Ser-46 by PGA_2_ treatment in HCT116 cells, and the extent of p53 phosphorylation was increased in parallel with concentrations of PGA_2_ (Figure 2A, Appendix A). Notably, the protein level of p53 was also increased in the same pattern with that of p53 phosphorylation, implying that phosphorylation of p53 protein may result in its stabilization. 

Transcripts of p53’s target genes such as *PUMA*, *BAX*, *p21^WAF1^*, and *NOXA* were augmented by PGA_2_ treatment in HCT116 cells but not in HCT116 p53-/- cells, suggesting that PGA_2_ should increase transcriptional activity of p53 by phosphorylating it (Figure 2B, Appendix A). Messenger RNAs of *p21^WAF1^* and *NOXA* showed an increasing tendency in PGA_2_–treated HCT116 p53 -/- cells, implying p53-independent gene expression by PGA_2_. However, pifithrin (PFT)-α, a chemical inhibitor of p53’s transcriptional activity, prevented the increase of *p21^WAF1^* and *NOXA* mRNAs in HCT116 cells, but not in HCT116 p53 -/- cells (Appendix A). It was reported that PGA_2_ can increase the level of *p21^WAF1^* mRNA via HuR-mediated stabilization of *p21^WAF1^* mRNA in the absence of p53 [23]. In addition, it was shown that transcription factors such as SP1, p73, ATF3 and ATF4 can activate transcription of NOXA without involvement of p53 [24,25]. So, it can be speculated that PGA_2_ activates p53-dependent transcription in HCT116 cells, and PGA_2_ also increases of *p21^WAF1^* and NOXA mRNAs in HCT116 p53-/- cells through mRNA stabilization and activation of various transcription factors. The increase of p21^WAF1^, PUMA, and NOXA expression was also observed at the level of protein (Figure 2C, Appendix A). While multiple protein kinases including HIPK-2, p38MAPK, PKC-δ, and DYRK2 phosphorylate p53 at Ser-46, enzymes involved in DNA damage response such as DNA-PK, ATM, and ATR, phosphorylate p53 at Ser-15 [26,27]. Since phosphorylation of p53 Ser-46 is very subtle and PGA_2_ was reported to activate DNA-PK [21], DNA-PK was suspected to be a principal enzyme to phosphorylate p53 Ser-15 and induce apoptosis. And as expected, NU7441, an inhibitor of DNA-activated protein kinase catalytic subunit (DNA-PKcs) prevented PGA_2_-induced phosphorylation of p53 and increase of p21^WAF1^ protein (Figure 2D), demonstrating a causative role of DNA-PK in activation of p53. To be consistent with its effect on PGA_2_-induced transcriptional activity of p53, NU7441 suppressed PGA_2_-induced apoptotic findings such as cleavage of PARP1 and increase of annexin V-positive cells as well (Figure 2D,E). NU7441 showed no effect on survival of HCT116 p53-/- cells (Appendix A). Accordingly, these data suggested that PGA_2_ may activate p53 via inducing DNA-PKcs activity, which plays a critical role in PGA_2_-induced apoptosis.

### 3.3. PGA_2_-Induced Apoptosis Was Dependent on the Transcriptional Activity of p53

Activation of p53 induces apoptosis in a manner of both dependent on and independent of its transcriptional activity in which these two pathways are not mutually exclusive [1,28]. Then, we attempted to determine whether the effect of p53 was exerted via transcriptional regulation of apoptotic genes or via its effect on the mitochondrial apoptotic pathway in PGA_2_-induced apoptosis. As shown in Figure 3A,B, and Appendix A, PGA_2_-induced apoptosis was gradually decreased according to concentrations of PFT-α. Consistently with these findings, cleavage of PARP1 and caspase-3 was reduced, but phosphorylation of p53 was marginally affected, indicating that the caspase activation cascade was downstream of p53 activation (Figure 3C). Considering that pretreatment of PFT-α reduced expression of PGA_2_-induced p21^WAF1^, the inhibitory effect of PFT-α on PGA_2_-induced apoptosis must be due to its inhibitory effect against the transcriptional activity of p53 (Figure 3C). Besides, p53 did not move to mitochondria and, instead, was accumulated in cytosol (Appendix A). Furthermore, phosphorylated p53, which has transcriptional activity, was increased in the nuclear fraction (Appendix A). Therefore, these results suggested that PGA_2_-induced apoptosis may be dependent on the transcriptional activity of p53 but not mitochondrial p53. 

### 3.4. PGA_2_-Induced Apoptosis Is Dependent on de novo Protein Synthesis of p53 Target Genes

Then, we speculated that proteins of p53 target genes might play a critical role in PGA_2_-induced apoptosis. To prove this speculation, we analyzed the effect of cycloheximide (CHX), an inhibitor of translation, on PGA_2_-induced apoptosis. Pretreatment of CHX prevented the induction of apoptosis by PGA_2_ (Appendix A), and cleavage of both PARP1 and caspase-3 as well without an effect on phosphorylation of p53 (Appendix A), implying the critical role of de novo proteins synthesized by p53 in this apoptosis model. Although PUMA and NOXA proteins were synthesized by p53, knockdown of PUMA and NOXA using siRNAs did not suppress PGA_2_-induced apoptosis (Appendix A). Not only pro-apoptotic BCL-2 family proteins but also death receptor (DR) proteins such as DR4 (tumor necrosis factor receptor superfamily member 10a, TNFRSF10A), DR5 (tumor necrosis factor receptor superfamily member 10b, TNFRSF10B), and FAS (Fas cell surface death receptor) are involved in the induction of p53-induced apoptosis [29,30]. Among death receptors, DR5 but neither DR4 nor FAS was significantly increased at the level of protein by PGA_2_, which was accompanied by phosphorylation of p53, an increase of p21^WAF1^ protein, and cleavage of both PARP1 and caspase-3 (Figure 4A). In the reporter-luciferase gene assay, the promoter activity of *DR5* gene was upregulated by PGA_2_ treatment in HCT116 cells but not in HCT116 p53-/- cells, which was suppressed by pretreatment with PFT-α, indicating an increase of *DR5* expression at the level of transcription by PGA_2_-induced transcriptional activity of p53 (Figure 4E). 

Supporting the result of reporter gene assay, expression of *DR5* mRNA was increased by PGA_2_ treatment, and this increase was alleviated by PFT-α pretreatment (Appendix A). Notably, knockdown of *DR5* expression using siRNA suppressed both an increase of annexin V-positive cells (Figure 4B) and cleavage of PARP1 (Figure 4C) in PGA_2_-treated cells. Moreover, both NU7441 and PFT-α, which suppressed PGA_2_-induced apoptosis, prevented the expression of DR5 as well as cleavage of PARP1 (Figure 4D). Collectively, these data suggested that DR5 upregulated by p53 plays a pivotal role in PGA_2_-induced apoptosis.

## 4. Discussion

The results of this study can be, collectively, summarized that PGA_2_ induces the activation of p53 in HCT116 cells via DNA-PK, and p53, in turn, upregulates the expression of *DR5* at the level of transcription, which finally leads to caspase-dependent apoptosis.

In this study, how PGA_2_ increases the activity of DNA-PKcs was not clarified. The activation of DNA-PKcs occurs through multiple pathways [31]. DNA-PKcs is primarily activated in response to the damaged DNA via interaction with Ku70 and Ku80 proteins, which are recruited to the end of broken DNA. DNA-PKcs is also activated through protein kinases such as AKT and casein kinase II even in the absence of DNA damage. In the previous report, PGA_2_-induced DNA-PKcs activity was owing to reactive oxygen species (ROS)-induced DNA damage [21]. However, 15d-PGJ_2_, another cyclopentenone PG, induced activation of ATM, a protein kinase responsible for breakage of double stranded DNA, by direct interaction with ATM at cysteine residues, resulting in activation of ATM [32]. So, it may be possible that PGA_2_ induces activation of DNA-PKcs through damaging DNA via ROS accumulation and physical interaction with DNA-PKcs as well. To identify the involvement of DNA damage in PGA_2_-induced apoptosis, phosphorylation of histone H2A variant H2AX at Ser-139 (γ-H2AX), which is a sensitive marker of DNA damage [33], was detected by immunoblot analysis. As shown in Appendix A, PGA_2_ treatment increased γ-H2AX remarkably, indicating that DNA damage occurred in PGA_2_-treated cells. Besides, pretreatment of PFT-α did not affect the level of γ-H2AX, while it reduced the level of cleaved PARP1 (Appendix A, Figure 4D). Moreover, PGA_2_ treatment increased γ-H2AX in HCT116 p53-/- cells as well, which was not suppressed by either PFT-α or NU7441 (Appendix A). Therefore, it can be speculated that activation of DNA-PK may be due to damaged DNA, and both DNA damage and hence activation of DNA-PK occur upstream of p53 activation and induction of apoptosis.

Many target genes of p53 are involved in apoptosis [29,30,31,34]. As shown in Appendix A, mRNA levels of p53 target genes that were reported to induce apoptosis including *FAS* and *FAS ligand (Fas-L)* in PGA_2_-treated cells were not significantly increased by PGA_2_ treatment. So, it seems that in HCT116 cells, PGA_2_-activated p53 induces expression of limited numbers of pro-apoptotic genes such as *BAX*, *PUMA*, *NOXA*, and *DR5*. Therefore, DNA-PK-p53-DR5 pathway might be a sole apoptosis-activating mechanism for PGA_2_-induced apoptosis in HCT116, although transcriptional target genes of p53 involved in apoptosis were not investigated at the level of transcriptome in this study, and thus the possibility of involvement of (a) novel gene(s) still remains. It is not clarified in this study how DNA-PK/p53 activation leads to specific transcriptional induction of DR5. Based on the report of Woo *et al*, it can be speculated that DNA-PK activation is necessary but not sufficient for p53-mediated DR5 transcription [35]. When HCT116 cells were treated with nutlin-3, an inhibitor of MDM2, which does not activate DNA-PK, p53 target genes were induced (Appendix A) and moreover, transcription of DR5 was turned out to increase through stabilized p53 [36]. Thus, DNA-PK activity seems to be necessary for stabilization of p53 but not binding of p53 protein to p53 response element in promoter region of target genes. 

Active p53 protein can affect both intrinsic and extrinsic apoptosis by upregulating the expression of several genes of each pathway or physical interaction with mitochondria [28]. Although it is not clear if p53 affect the intrinsic apoptosis or the extrinsic apoptosis at this point in PGA_2_-induced apoptosis in HCT116 cells, considering no attenuating effect of siRNAs against *PUMA* and *NOXA* on PGA_2_-induced apoptosis (Appendix A) and no release of cytochrome c from mitochondria into cytosol (Appendix A) leads to the speculation that PGA_2_-induced apoptosis in HCT116 cells may occur through the extrinsic apoptotic pathway which is activated by upregulated DR5.

Several reports have demonstrated that PGA_2_ inhibits the growth and induce cell death in cancer cells [8,37]. Although the growth inhibitory pathway and apoptotic pathway induced by PGA_2_ are all dependent on de novo protein synthesis, their relationship is not clearly established. In this study, PGA_2_ induced apoptosis in HCT116 cells containing wild-type p53 gene to a much higher extent than that in HCT116 p53-/- cells. However, surprisingly, PGA_2_ inhibited the growth of both HCT116 cells and HCT116 p53-/- cells to similar extent in a dose-dependent manner by CCK-8 assay which detects live cells (Appendix A). In cell cycle distribution analysis, PGA_2_ induced G2M arrest in both cell lines, but accumulated sub-G1 apoptotic cells, only in HCT116 cells (Appendix A). Therefore, the growth inhibitory effect of PGA_2_ may be exerted through both p53-dependent and –independent manners in HCT116 cells, although PGA_2_-induced apoptosis was dependent on p53 in HCT116 cells. Based on the result that PGA_2_ induced G2M arrest at higher level in HCT116 p53 -/- cells than in HCT116 cell, it can be assumed that PGA_2_ might activate stronger cell cycle arrest in HCT116 p53 -/- cells than that in HCT116 cells or PGA_2_ might activate p53-independent growth inhibitory mechanism in HCT116 p53 -/- cells, finally resulting in similar CCK-8 results. 

Besides, PGA_2_ has also been reported to induce apoptosis in HL-60 cells and Hep3B cells whose p53 is deleted or mutated [11,15]. So, the dependency of PGA_2_-induced apoptosis on p53 may not be applied for all cell types. In HL-60 cells, PGA_2_ induced intrinsic apoptosis by direct interaction with mitochondria without the involvement of de novo protein synthesis. In Hep3B cells, PGA_2_ induced apoptosis via up-regulation of SOX-4, suggesting that PGA_2_ can induce apoptosis via multiple pathways according to cellular contexts. Interestingly, PGA_2_ induced apoptosis in HepG2 and Hep3B cells in a caspase-independent manner. Therefore, these reports suggested that PGA_2_ induces apoptosis via the combinatorial pathways composed of caspase and new protein synthesis. The involvement of caspase activity may be determined by the proteins induced by PGA_2_. 

Since cyclooxygenase-2 (COX-2) which synthesizes PGE_2_ is highly expressed in many types of cancer, PGE_2_ is elevated in cancer tissues. PGE_2_ secreted from cancer tissues increases cancer cell survival, angiogenesis, invasion, and metastasis, playing a promoting role in carcinogenesis and immunosuppression in microenvironment around cancers [38,39]. Thus, COX-2/PGE_2_ axis has been proposed as a therapeutic target in cancer treatment [40]. In the meanwhile, PGA_2_ which is produced from PGE_2_ by non-enzymatic dehydration, shows opposite effects to those of PGE_2_ against cancers, proposing PGA_2_ as a therapeutic molecule for cancer treatment [41]. So, it can be speculated that the relative amount between PGE_2_ and PGA_2_ in tumor microenvironment may contribute to the determination of the efficacy of both anti-cancer chemotherapy and radiotherapy, the effect of which is dependent on p53 activity. Therefore, the genetic information of cancer tissues should be integrated with profiles of inflammatory cytokines, including PGE_2_ and PGA_2_ in tumor microenvironment to maximize the effects of anti-cancer therapeutics in the future.

## 5. Conclusions

Treatment of PGA_2_ induces caspase-dependent apoptosis in HCT116 cells containing wild type p53 gene, but not in HCT116 p53-/- cells which are deficient in functional p53. Activation of p53 by PGA_2_ in HCT116 cells was dependent on the activity of DNA-PKcs. Among transcriptional target genes of p53, *DR5* was responsible for the induction of apoptosis in PGA_2_-treated cells. PGA_2_ increases the expression of *DR5* at the level of transcription via p53 activity. Therefore, PGA_2_ induces p53-dependent apoptosis by activating DNA-PKcs-p53-DR5 pathway in HCT116 colorectal cancer cells. 

## Figures and Tables

**Figure 1 biomolecules-10-00492-f001:**
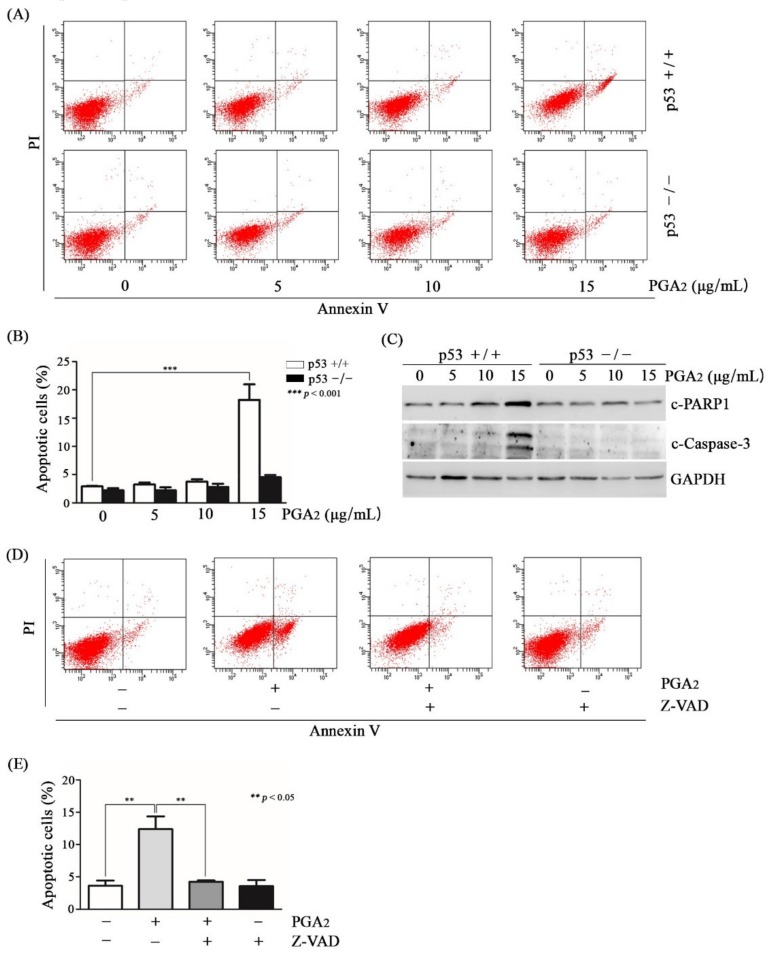
Comparison of PGA_2_-induced apoptosis between HCT116 cells (p53 +/+) and HCT116 p53 null cells (p53 -/-). (**A**) Cells were seeded at a density of 4 x 10^5^ cells/well and treated with indicated concentrations of PGA_2_ the next day. After 12 h post-treatment, cells were stained with annexin V and propidium iodide, which were subjected to flow cytometric analysis. The result is representative of three independent experiments. (**B**) The result of three independent annexin V assay performed in (**A**) was presented as mean ± standard error of the mean (SEM). (**C**) Whole cell lysates (WCL) of two cell lines treated the same as described in (**A**) were subjected to immunoblot analysis against cleaved PARP1 (c-PARP1), cleaved caspase-3 (c-Caspase-3), and glyceraldehyde-3-phosphate dehydrogenase (GAPDH) which was used as an internal reference protein for normalization. (**D**) HCT116 cells were pretreated with z-VAD-Fmk for 1 h and treated with PGA_2_ (15 µg/mL) for another 12 h. Cells were then subjected to annexin V assay. The result is representative of three independent experiments. (**E**) The result of three independent annexin V assay performed in (**D**) was presented as mean ± SEM.

**Figure 2 biomolecules-10-00492-f002:**
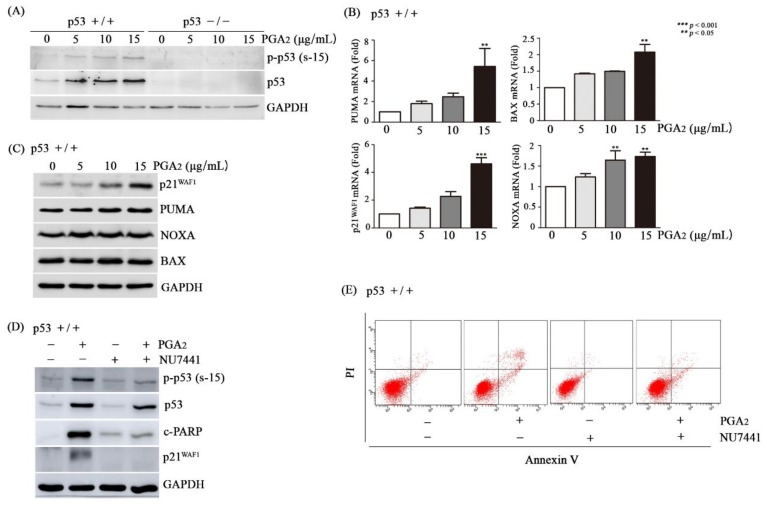
Activation of p53 by DNA-PK in PGA_2_-treated HCT116 cells. (**A**) HCT116 cells (p53 +/+) and HCT116 p53 null cells (p53 -/-) were treated with indicated concentrations of PGA_2_. After 12 h, WCLs were subjected to immunoblot analysis against phospho-p53 [Ser-15, p-p53 (s-15)], p53, and GAPDH as an internal reference protein. (**B**) Total cellular RNA of HCT116 cells treated with indicated concentrations of PGA_2_ for 12 h were subjected to qPCR against indicated genes using *GADPH* as an internal reference gene for normalization. (**C**) HCT116 cells treated the same as described in (**B**) were subjected to immunoblot analysis against indicated proteins using GADPH as an internal reference protein. (**D**) HCT116 cells were treated with vehicle or NU7441, an inhibitor of DNA-PK, for 1 h and were incubated in the absence or presence of PGA_2_ (15 µg/mL) for another 18 h. Cells were then subjected to immunoblot analysis against phospho-p53 (Ser-15), p53, p21^WAF1^, cleaved PARP1 (c-PARP1), and cleaved caspase-3 (c-Caspase-3) using GAPDH as an internal reference protein. (**E**) HCT116 cells treated the same as described in (**D**) were subjected to annexin V assay. The result is representative of three independent experiments.

**Figure 3 biomolecules-10-00492-f003:**
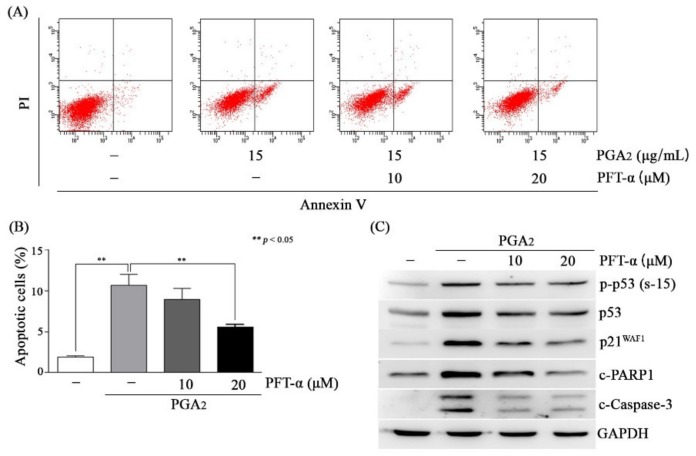
The effect of pifithrin-α (PFT-α) on PGA_2_-induced apoptosis in HCT116 cells. (**A**,**B**) HCT116 cells treated with indicated concentrations of PFT-α for 1 h were incubated in the absence or presence of PGA_2_ (15 µg/mL) for another 18 h. Cells were then subjected to annexin V assay. The representative images and statistical analysis of three independent experiments were shown in (**A**) and (**B**), respectively. The result of three independent annexin V assay was presented as mean ± SEM. (**C**) Whole cell lysates (WCLs) were prepared and subjected to immunoblot analysis against indicated proteins using GAPDH as an internal reference protein.

**Figure 4 biomolecules-10-00492-f004:**
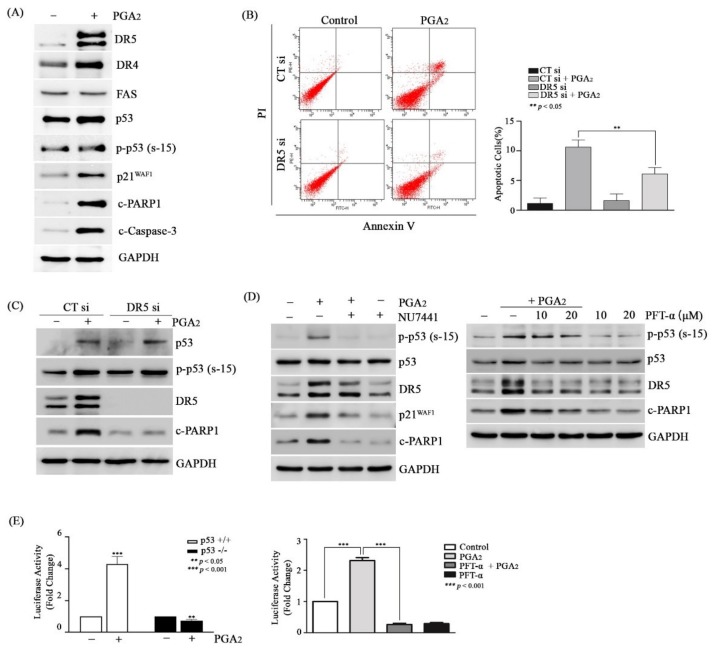
The effect of DR5 on PGA_2_-induced apoptosis in HCT116 cells. (**A**) HCT116 cells were treated with PGA_2_ (15 µg/mL). After 18 h, WCLs were subjected to immunoblot analysis against indicated proteins using GAPDH as an internal reference protein. (**B**) HCT116 cells were transfected with scrambled RNA or siRNA targeting *DR5* for 24 h and incubated in the presence of vehicle or PGA_2_ (15 µg/ml) for an additional 18 h. Cells were harvested and subjected to annexin V assay. The representative images (left) and statistical analysis (right) of three independent experiments were shown, respectively. The result of three independent annexin V assay was presented as mean ± SEM. (**C**) WCLs of HCT116 cells treated the same as described in (**B**) were subjected to immunoblot analysis against indicated proteins using GAPDH as an internal reference protein. (**D**) HCT116 cells treated with Nu-7441 (left) or PFT-α (right) for 1 h were incubated in the presence of vehicle or PGA_2_ (15 µg/ml) for another 18 h. WCLs were subjected to immunoblot analysis against indicated proteins using GADPH as an internal reference protein. (**E**) (left) After DR5 promoter-luciferase construct (pGL3-DR5) was transfected into HCT116 cells and HCT116 p53-/- cells along with renilla luciferase for 24 h, HCT116 cells were treated with vehicle or PGA_2_ (15 µg/mL). (right) After DR5 promoter-luciferase construct (pGL3-DR5) was transfected into HCT116 cells along with renilla luciferase for 24 h, HCT116 cells were treated with vehicle or PFT-α (20 µM) for 1 h and were incubated in the absence or presence of PGA_2_ (15 µg/mL). At 18 h post-treatment of PGA_2_, firefly luciferase activity of DR5 promoter was measured by Dual luciferase assay method using renilla luciferase activity as the normalizer.

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
