# Peer review of "Induction of p53-Dependent Apoptosis by Prostaglandin A2"

_biomolecules, 2020, doi:10.3390/biom10030492_

Round 1
Reviewer 1 Report
I have no concern about the experiments of this study, and I appreciate the efforts of the authors to answer all reviewers’ questions. My concern is that the main message of this study is not correct.
The authors in the abstract mention:
PGA2 induced cell death associated with activation of caspase‐3 and cleavage of poly(ADP‐ribose) polymerase 1 in HCT116 cells, but it did not induce these phenomena in HCT116 p53‐/‐ cells.
I think the last sentence is not correct as PGA appears to induce cell apoptosis also in HCTp53-/- although with a time delay and to a less extent than p53+/+.
This appears from the following figures (and from authors’ comments):
- In Figure S1A, C, it appears that cell apoptosis also occurs in HCT p53-/- although delayed (at day 3 instead of day 2) and with reduced levels compared to p53+/+.
- Similarly, in Fig S13 (now 14), there is an evident cell viability reduction. This cannot be commented as growth inhibition, because it is a decrease of cell number (not a steady-state).
Moreover, as the authors mention and comment, PGA is active on HCTp53-/-, by inhibiting growth:
- “in Figure S14 (Figure S13 in the old version) which shows clearly a growth inhibitory effect of PGA2 in a dose-dependent manner, the growth inhibitory effect was similar in both cell lines, although the growth inhibitory effect of PGA2 is still a little stronger in HCT116 cells than in HCT116 p53-/- cells”.
- “As shown in Figure S10, PGA2 induces DNA damage common in both cells”.
- Therefore, although we did not analyze the mechanisms of cell cycle arrest in PGA2-treated cells, the growth inhibitory effect of PGA2 between the two cell lines may not differ by the degree of apoptosis.
I think that the authors demonstrate that p53 is one of the players of PGA-induced apoptosis but not the only one. This is important from the application point of view.
So, my final advice is that the authors should comment more appropriately their findings in the abstract.
Reviewer 2 Report
In this manuscript the authors evaluated the effect of Prostaglandin (PG) A2 on apoptosis in colon cancer cell line (HCT116) with or without endogenous wtp53. They found that PGA2 induced apoptosis and that p53 played a role.
The study is interesting, although many other similar studies have been published, and few amendments need to be done to improve the manuscript.
In the results section the authors write a mix a results and discussion, please just write the results as they appear in the figures and move the comments in the discussion part.
The discussion section, on the other hand, is unnecessarily too long, please try to improve it.
Author Response
Please see the attachment

This manuscript is a resubmission of an earlier submission. The following is a list of the peer review reports and author responses from that submission.
Round 1
Reviewer 1 Report
In this manuscript the authors evaluated the effect of Prostaglandin (PG) A2 on apoptosis in colon cancer cell line (HCT116) with or without endogenous wtp53. They found that PGA2 induced apoptosis and that p53 played a role.
The study is quite limited, since many other similar studies have been published, however it could be improved by adding some significant experiments.
Major points:
- The authors claim that p53 is activated because its phosphorylation status after PGA2 treatment. However, the authors need to analyse at least p-Ser46-p53, because it is the phosphorylated residue implicated in p53 apoptotic activation (D’Orazi et al., Nat Cell Biol. 2002 Jan;4(1):11-9. PMID:11780126).
- The authors should also perform a time course after PGA2 treatment (24-48 72 h) since they show higher expression of p21 which usually induces cell cycle arrest. Increased apoptotic markers could be detected after longer exposure to PGA2.
- The authors should demonstrate the mechanism of p53 activation by DNA-PK. In this regard, the authors should assess by WB the DNA damage, by using for instance antibody anti-p-H2AX.
- They should also add the reference that showed that “DNA-PK is necessary but not sufficient for activation of p53 sequence-specific DNA binding, and that this activation occurs in response to DNA damage”, Nature. 1998 Aug 13;394(6694):700-4.
Minor points:
- In the abstract I would change “the effect of p53” with “the role of p53”.
In the discussion section the authors claim that :” Many target genes of p53 are involved in apoptosis [24‐26,28]. Since other genes than DR5 may possibly be involved in PGA2‐induced apoptosis, we examined the expression change of p53 target genes that were reported to induce apoptosis including FAS, FAS‐L, insulin‐like growth factor binding protein 3 (IGFBP3), glycosylphosphatidylinositol‐ anchored molecule like (GML), purinergic receptor P2X 6 (P2RX6), p53 apoptosis effector related to PMP22 (PERP), zinc finger matrin‐type 3 (ZMAT3), p53‐induced death domain protein 1 (PIDD), apoptotic peptidase activating factor 1 (APAF1), caspase‐6, BCL2 interacting killer (BIK), and BCL2 associated agonist of cell death (BAD) in PGA2‐treated cells”, however they are not shown in the text. Please explain andamend.
- In the Introduction and Discusion sections the authors should better describe the implication of PGA2 in cancer: When is PGA2 found? Why? Which is the physiological role of PGA2? Why PGA2 it should be relevant for p53 activation? In which phusiopathological status?
Author Response
In this manuscript the authors evaluated the effect of Prostaglandin (PG) A2 on apoptosis in colon cancer cell line (HCT116) with or without endogenous wtp53. They found that PGA2 induced apoptosis and that p53 played a role.
The study is quite limited, since many other similar studies have been published, however it could be improved by adding some significant experiments.
Answer: I deeply appreciate your positive comment on the manuscript. I believe that your suggestion to add some experimental data will definitely make the manuscript more complete.
Major points:
- The authors claim that p53 is activated because its phosphorylation status after PGA2 treatment. However, the authors need to analyse at least p-Ser46-p53, because it is the phosphorylated residue implicated in p53 apoptotic activation (D’Orazi et al., Nat Cell Biol. 2002 Jan;4(1):11-9. PMID:11780126).
Thank you so much for your very valuable comment. Following your suggestion, we analyze the phosphorylation of p-Ser46-p53. As shown in Supplementary Figure S2, p-Ser46-p53 was phosphorylated according to the concentrations of PGA2. I think this data may support the conclusion of this manuscript that PGA2 induces apoptosis mediated by activation of p53 once again.
- The authors should also perform a time course after PGA2 treatment (24-48 72 h) since they show higher expression of p21 which usually induces cell cycle arrest. Increased apoptotic markers could be detected after longer exposure to PGA2.
Thank you for your comment. Per your suggestion, we analyze the effect of PGA2 on the growth of HCT116 cells for 48 and 72 hours using annexin v assay and gene expression analysis such as qPCR and western blot. Since 15 and 20 ug/ml of PGA2 induced too much apoptosis to analyze after 2 and 3 days, we analyzed the cell death induced by 10 ug/ml of PGA2. As shown in Supplementary Figure S1, PGA2 (10 ug/ml) induced significantly apoptosis after 2 and 3 days, which was also prevented by PFT-a. Therefore, these results suggest that PGA2 induced p53-dependent apoptosis in a dose- and time-dependent manners in HCT116 cells.
In cell cycle analysis, PGA2 induced G2M arrest instead of G1 arrest in HCT116 cells (Supplementary Figure S14). Since p53 activates G2M arrest as well as G1 arrest, PGA2 treatment induces G2M arrest mediated by p53-p21WAF1 in HCT116 cells. Notably, PGA2 increases p21WAF1 mRNA in HCT116 p53-/- cells (Supplementary Figure S3). It was reported that PGA2 can increase the level of p21WAF1 mRNA via HuR-mediated stabilization of p21WAF1 mRNA in the absence of p53 [23]. So, increase of p21WAF1 mRNA in HCT116 p53-/- cells by PGA2 might be due to mRNA stabilization, which is irrespective of p53. So, even in HCT116 p53-/- cells, it seems that PGA2 induces G2M arrest mediated by p21WAF1. Therefore, although the apoptosis-inducing activity of PGA2 is dependent on p53, the growth suppressive effect of PGA2 on HCT116 cells can be exerted through p53-dependent and –independent manners in HCT116 cells as shown in CCK-8 assay (Supplementary Figure S13).
- The authors should demonstrate the mechanism of p53 activation by DNA-PK. In this regard, the authors should assess by WB the DNA damage, by using for instance antibody anti-p-H2AX.
Thank you so much for the valuable comment. Per your comment, we analyzed the expression of phosphor-H2AX in this model. As shown in Supplementary Figure S9, PGA2 treatment increased expression of phospho-H2AX both in HCT116 and HCT116 p53-/- cells, indicating that PGA2 induced DNA damage in HCT116 cells. Moreover, the increased expression of phospho-H2AX was not attenuated by PFT-a and NU7441 pretreatment. Therefore, these findings suggest that PGA2 treatment should induce DNA damage, which leads to phosphorylation and activation of p53, and finally apoptosis.
- They should also add the reference that showed that “DNA-PK is necessary but not sufficient for activation of p53 sequence-specific DNA binding, and that this activation occurs in response to DNA damage”, Nature. 1998 Aug 13;394(6694):700-4.
Thank you for your nice comment and suggestion. Following your suggestion, we added the reference and paragraph in the text as follows.
It is not clarified in this study how DNA-PK/p53 activation leads to specific transcriptional induction of DR5. Based on the report of Woo et al, it can be speculated that DNA-PK activation is necessary but not sufficient for p53-mediated DR5 transcription [33]. When HCT116 cells were treated with nutlin-3, an inhibitor of MDM2, which does not activate DNA-PK, p53 target genes were induced (Figure S11) and moreover, transcription of DR5 was turned out to increase through stabilized p53 [34]. Thus, DNA-PK activity seems to be necessary for stabilization of p53 but not binding of p53 protein to p53 response element in promoter region of target genes.
Minor points:
- In the abstract I would change “the effect of p53” with “the role of p53”.
Thank you for correcting my mistake. I agree with you that “the effect of p53” should be changed to “the role of p53” and changed it.
In the discussion section the authors claim that :” Many target genes of p53 are involved in apoptosis [24‐26,28]. Since other genes than DR5 may possibly be involved in PGA2‐induced apoptosis, we examined the expression change of p53 target genes that were reported to induce apoptosis including FAS, FAS‐L, insulin‐like growth factor binding protein 3 (IGFBP3), glycosylphosphatidylinositol‐ anchored molecule like (GML), purinergic receptor P2X 6 (P2RX6), p53 apoptosis effector related to PMP22 (PERP), zinc finger matrin‐type 3 (ZMAT3), p53‐induced death domain protein 1 (PIDD), apoptotic peptidase activating factor 1 (APAF1), caspase‐6, BCL2 interacting killer (BIK), and BCL2 associated agonist of cell death (BAD) in PGA2‐treated cells”, however they are not shown in the text. Please explain and amend.
Thank you so much for your critical comment and I feel sorry to make you confused. Per your comment, we added qPCR data of listed p53 target genes (Supplementary Figure S10). As shown in this data, p53 target genes that were reported to mediate the p53-induced apoptosis were not elevated by PGA2 treatment. Thus, PGA2-induced apoptosis may be solely dependent on DR5 which is elevated by p53.
- In the Introduction and Discusion sections the authors should better describe the implication of PGA2 in cancer: When is PGA2 found? Why? Which is the physiological role of PGA2? Why PGA2 it should be relevant for p53 activation? In which phusiopathological status?
Thank you for the excellent comment. PGA2 is one of the cyclopentenone PGs and is produced from PGE2 by non-enzymatic dehydration process. Cyclopenteone PGs have various biological functions including anti-inflammatory effect, growth inhibitory effect on cancer cells, and anti-viral effects. The functions of cyclopentenone PGs are almost opposite to those of cyclopentane PGs such as PGE2. These phenomena have been described in many journals. So, we described pathophysiological aspect of PGA2 more in the Discussion as follows.
Since cyclooxygenase-2 (COX-2) which synthesizes PGE2 is highly expressed in many types of cancer, PGE2 is elevated in cancer tissues. PGE2 secreted from cancer tissues increases cancer cell survival, angiogenesis, invasion, and metastasis, playing a promoting role in carcinogenesis and immunosuppression in microenvironment around cancers [29,30]. Thus, COX-2/PGE2 axis has been proposed as a therapeutic target in cancer treatment [38]. In the meanwhile, PGA2 which is produced from PGE2 by non-enzymatic dehydration, against cancers shows opposite effects to those of PGE2, proposing PGA2 as a therapeutic molecule for cancer treatment [39]. So, it can be speculated that the relative amount between PGE2 and PGA2 in tumor microenvironment may contribute to the determination of the efficacy of both anti-cancer chemotherapy and radiotherapy, the effect of which is dependent on p53 activity. Therefore, the genetic information of cancer tissues should be integrated with profiles of inflammatory cytokines, including PGE2 and PGA2 in tumor microenvironment to maximize the effects of anti-cancer therapeutics in the future.
Reviewer 2 Report
The authors aim to demonstrate the relationship between PGa2 and p53, particularly in mediating cell apoptosis. The main problem of this paper is the lack of control: the use of inhibitors, as well as the check of p53 target genes, should also be verified in the absence of p53 to demonstrate the requirement of p53 in PGA2-mediated apoptosis. Furthermore, while the effects of PGA2 on p53 are dose-dependent, the effects on cell growth are not, raising doubt about the p53/PGA2 relationship.
Major revisions:
The PGA2 effects on cell apoptosis are visible only at 15ug/ml, whereas they are completely absent at doses 5-10. The lack of a dose-response relationship is anomalous. The authors need to verify these effects at higher doses. The authors should verify the specificity of P53 post-translational modification (PTMs) by using molecules known to induce p53 but not PTMs (i.e. Nutlin). Additionally, others PTMs mediate p53 apoptotic function (as ser46). The authors need to verify them Transcripts of p53’s target genes such as PUMA, BAX, p21WAF1, and NOXA should also be analysed in p53-/- cells to ascertain the specificity of p53 activity. Moreover, the quantification of protein levels should be performed. (The increase of Noxa levels is hard to detect). The effects of Nu7741 should be verified also on HCT116 p53-/-as well as that of PFTa. As demonstrated in figure 4d, Nu7741 is effective also in the absence of PGA2. So the p53 role is not demonstrated. P53-mediated apoptosis transcription-dependent and independent are not mutually exclusive. The authors should correct the sentence. Authors should show all “data not shown”. The promoter activity of DR5 gene should be ascertained in p53-/- cells.
Round 2
Reviewer 1 Report
The authors satisfactorily addressed the reviewer's comments
Reviewer 2 Report
The authors have performed new experiments as requested by the reviewer. However, I think the new experiments do not completely support the authors conclusions: they indicate that PGA2 induces some p53-targets genes also in HCT116 p53-/- (p21 and Noxa) as recognized also by the authors. The dose-response of PGA2 on cell apoptosis is not convincing: at the dose of 20, the main response is necrosis and not apoptosis (how did the authors calculate cell apoptosis from FACS picture?
Importantly, Supplementary Fig S13 demonstrates that cell viability is similarly affected by PGA2 in HCTR116 p53-/- and +/+. This is somehow surprising: how can a substance induce apoptosis only in the presence of p53+/+, whereas cell viability (that derives from cell proliferation plus cell apoptosis) is the same in presence or absence of p53?
The text of the manuscript was not changed accordingly to the new experiments and the majority of the supplementary figures are not cited or are not correctly cited.